# The Impact of Household Wealth on Adoption and Compliance to GLOBAL GAP Production Standards: Evidence from Smallholder farmers in Kenya

**Castro N. Gichuki [1], Jiqin Han [1,\*] and Tim Njagi [2]**

[1]  College of Economics and Management, China Center for Food Security Studies,
    Nanjing Agricultural University, Nanjing 210095, China; ngumbucg@gmail.com
[2]  Tegemeo Institute of Agricultural Policy and Development, Research, Egerton University,
    Nairobi 20498-00200, Kenya; tnjagi@tegemeo.org
\*  Correspondence: jhan@njau.edu.cn; Tel.: +86-13815-875-653

**Abstract:** Horticultural production for the export market has been credited for rural community growth and employment in Sub-Saharan Africa. To make the agri-enterprises competitive and profitable, smallholder farmers are now required to invest in food safety and production standards such as GLOBAL Good Agricultural Practices (GLOBAL GAP). Most often, the inability to afford capital has affected compliance with food safety production standards. However, farmers can use household assets to finance the agri enterprise ventures. The purpose of the study is to explore the impact of household wealth on the adoption of food safety certification standards. The study used cross-sectional data collected from 479 smallholder farmers in Kenya. The findings reveal that 49% of the households are categorized as wealthy and 51% as poorly endowed. The empirical findings on the willingness to adopt GLOBAL GAP certification reveal that membership to GLOBAL GAP affiliated farmers groups significantly influence on wealthier farmers to attain GLOBAL GAP certification status. Farmers groups facilitate joint investments that reduce the cost of investing in GLOBAL GAP assets such as grading shed, protective gear, shower rooms, disposal pits, incinerators, hessian coolers, packaging crates, soil testing kits and establishing food traceability systems. In comparison to poor endowed farming households, the well-endowed farmers have a relatively significantly better wealth index, an indication that they can easily raise capital investments to finance GAP certification. Finally, we observe that selling beans to GAP certified buyers significantly enables farmers to comply with the expected certified production standards.

**Keywords:** food safety; household wealth index; smallholder farmers; GLOBAL GAP certification

## 1. Introduction

In developing countries, horticultural production for the export market has been credited for rural community growth and employment. In Sub-Saharan African countries, the export horticultural crops have become major foreign exchange-earners replacing traditional crops such as coffee, tea, pyrethrumHumphrey [1]. To increase the productivity and marketability of the crops, policies that emphasize food safety and quality standards along the chain have been developed. Also, market liberalization has increased the role of private agencies in coordinating and supervising food safety standards for the export market. Previous studies have relatively focused on the impact of food safety standards in enhancing the agricultural competitiveness of farmers in developing countries. The standards provide clear organization and management of the export supply chains and regulations of the horticultural sector [2].

Further, food standards assist farmers to break poverty traps by guaranteeing the ready market and high premiums returns. However, smallholder farmers producing for export markets have been confronted with stringent food safety requirements. Producing under food safety guideline has been linked to low prices that do not compensate for the investments [3]. Further, food standards have hidden health costs that farmers incur indirectly from regularly using pesticides [4].

To make the agri-enterprises more profitable, smallholder farmers must be willing to invest in new agricultural technologies and food certification standards. Conventionally, lack of liquid assets to pay for profit-enhancing technologies upfront and lack of access to credit has hampered the adoption of new agricultural adoption by farmers in developing countries [5]. Theoretically, the nature of the imperfect credit market assumes that only farmers with collateral would secure credit to finance new agricultural technologies. The need to increase productivity and returns lead farmers to sell productive assets such as bicycles, motorcycles, radios, etc. to raise funds for new technologies [6]. However, irrespective of wealth status, adoption levels of technologies vary based on the social and economic setups of the community [7]. When classifying different rural households based on their overall wealth, higher wealth status does not automatically qualify house households to be early adopters of innovations [8]. Generally, members of lower wealth strata may be willing to risk investing in new technologies because of the desire to improve their economic status. Empirically, there exists non-linearity between the wealth and adoption of technologies in Mexico [9].

Presently there is a wide array of literature that investigates the impact of social, economic status of rural farming households in developing countries. To sustain the needs and increase on-farm investments, household members of different age and sex groups engage in different ventures [10]. Specifically, rural households seek livelihood and diversification capital from off-farm employment, ownership of enterprises, and remittances from migrated household members [11,12]. However, the inability to manage accumulated farm resources may affect farmer's capacity to increase productivity and even meet sufficiency food needs.

Depending on the social, economic status, farmers tend to respond differently towards a lack of capital resources [13]. Wealthier farmers can easily secure financial capital than poor farmers who lack equitable assets and are regarded as less creditworthy. Regardless of access to financial capital challenges, farmers can use the accumulated household assets to finance ventures they perceive to be rewarding. As economic theory predicts, farmer's wealth status would greatly impact on the capacity to cope with both production risk, price risk and even investing in new technologies Langyintuo and Mungoma [14]. The relationship between wealth and adoption of innovation are mixed and complex, mainly because the distribution of productive assets vary between communities. Also, adoption of agricultural innovations is more likely linked to crop choice, crop income and not necessarily economic status of farmers [15].

The association between the use of innovations and farmer's wellbeing can be complicated. However, exogenous factors such as off-farm payments can impact negatively or positively on the association [16]. Even more important is understanding the complex relationship between household size, household production capacity, cultural practices, and accessibility to public infrastructure such as roads, markets, electricity [17]. While there exists a distinct relationship between assets and farmer's ability to invest in farm inputs, financial endowment by wealthier households would impact on the use of agricultural innovations [14]. In most of the cases, poorer farmers are unable to afford high-end innovations. On the contrary, farmers in the lower wealth quantum are more willing to risk in the adoption of innovation because of the desire to improve their wellbeing [8]. Generally, the literature does not explicitly show that household poverty does influence the adoption of innovations. Consequently, the ultimate goal of this study is to empirically show the impact of smallholder farmer's household wealth on adoption and compliance of food safety certification standards such as GLOBAL GAP. The remaining part of this paper is organized as follows. The second section previews the materials and methods used in the study, while results are presented in Section 3. The conclusion is presented in Section 4.

## 2. Materials and Methods

*2.1. Sampling Procedure*

The data used in the analysis was collected between September and October 2017 in Kirinyaga, Murang'a and Embu counties in Kenya (see Figure 1). The selected regions produce more than 60 percent of snap beans in Kenya. We used a multistage sampling procedure to select respondents. In the first stage, the purposive sampling method was used to cluster farmers in nine sub-counties drawn from the three counties (Kirinyaga, Murang'a and Embu). The second stage, random sampling technique to select farming households from nine sub-counties drawn from the three counties. A sample size of 479 farmers was selected from the population of snap bean smallholder farmers. The required sample size was determined by Yamane [18] sampling methodology formula.

$$n = \frac{N}{1 + Ne^2} \qquad (1)$$

where; $n$ = sample size (475), $N$ = Population (1 + 1200), while $e^2$ = Level of precision $(0.05)^2$. The identified farmers answered a detailed questionnaire on snap bean farming, management, production, harvesting, and marketing. Further, farmers were presented with probing questions on perceptions and knowledge awareness of GLOBAL GAP standards and certification process. The respondents provided information on household characteristics, assets, farm size, social capital, as well as non-income indicators. After data cleansing, the study worked with a sample of 450 farming households.

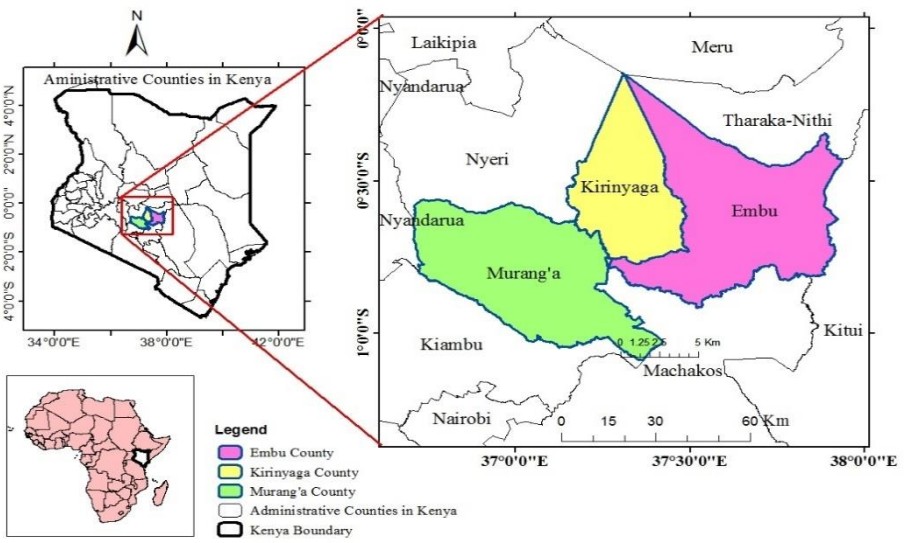

**Figure 1.** Map of the study area. Source: Authors' survey, 2017.

*2.2. Empirical Estimation*

2.2.1. Principle Component Analysis (PCA)

Previously, *PCA* technique has been used for compression and classification of data to reduce the dimensionality of the data set. Also, *PCA* creates a new data set by finding a new set of variables smaller than the original while retaining most of the sample information [19]. To investigate the potential impact of wealth on the adoption of GLOBAL GAPs certification, we first establish wealth indexes based on the farmer's household assets. Filmer and Pritchett [20] used PCA for the estimation of wealth levels using household assets indicators rather than income or consumption indicators traditionally used. This is mainly because the asset-based measurement can depict the household's long-run economic well-being and does not expressly account for short-term economic well-being and

economic shocks. Theoretically, the income variable is assumed to correlate with the given wealth levels, but this may not necessarily be the case as found by Gasparini, Sosa Escudero [21]. By the nature of the household set up in most of the developing countries, the productive assets owned by farmers significantly contribute to the wealth status but varies greatly between households [22]. When ranking households based on their economic status, it becomes more important to normalize (weights) the data on assets, this reduces or avoids data distortion resulting from different scales [16].

In expressing PCA model, the first principle component of a particular set of variables is considered to be linear index of all the variables and capturers the largest amount of information that is common among all the variables. Assuming we have a set of k assets by each farming household j. The each of the identified assets has to be normalized by standards division and its mean [14]. For instance, $a_{1j} = \left(a_{ij}^* - a_i^i\right)/s_i^*$, where $a_i^*$ the mean of $a_{1j}$ across all the farming households while $s_i^*$ is the standard deviation. The selected variables are expressed as linear combinations in a set of underlying components for every farming household j, expressed as;

$$a_{1j} = v_{11}A_{1j} + v_{12}A_{2j} + \ldots + v_{1k}A_{kj} \qquad \forall j = 1, \ldots\ldots, j.\ldots \tag{2}$$

$$a_{k1j} = v_{k1}A_{1j} + v_{k2}A_{zj} + \ldots v_{kk}A_{kj}$$

where the first principle component, expressed as the unnormalized variables, thus the index for each household can be expressed as

$$A_{1j} = \frac{f_{11}\left(a_{1j}^* - a_1^*\right)}{s_1^*} + \ldots + f_{1k}\left(a_{kj}^* - a_k^*\right)/\left(s_k^*\right) \tag{3}$$

The most critical assumption of PCA is that the undefined common information is in fact determined by the underlying common information that the index trying to estimate wealth levels. The first principle component variable across all the variables has a zero mean of which corresponds to the largest eigenvalues and correlates to the matrix. The first principle component gives a wealth index that assigns more weight to assets that vary across house households. This implies that assets found in most of the households are given zero weight.

### 2.2.2. Double Hurdle Model

The primary objective of this study is to explore the effects of smallholder farmer's household wealth on the adoption of GLOBAL GAP standards. The double-hurdle model is a twofold estimation designed to deal with survey data, which has many zero observations and continuous dependent variables [23]. The previous studies by Gao, Wailes [24] have used the double hurdle model estimation to estimate the adoption of agricultural technologies. While ordinary least squares can be considered, the high likelihood of biased results of the parameter estimates that do not take into account that the data is limited on one end. The bias would be more severe when the dependent variable is zero for a section of the data. In the case of analyzing for the adoption of technologies, the Two-step Heckman model could also be applied [25]. The correction of selection bias is applied to non-randomly selected samples and not randomly selected samples like in our case [26]. Therefore double hurdle is preferred for the study; the model is recommended by Jones [27]. The hurdle model is commonly associated with developing the econometric specification and integration into consumer choice theory.

The double hurdle model is estimated using maximum likelihood estimation technique. Before the model is estimated, it is necessary to overcome inconsistency arising from the presence of heteroscedasticity and non-normality of the error terms [28]. The required specification is made to allow for heteroscedasticity, this is established by letting the variance error to diverge across an observation by specifying a function of continuous variables set. Therefore, the analysis of standard deviation is specified as;

$$\sigma_i = exp(z_i'h) \tag{4}$$

where $z_i$ denote the elements of $x_i$. [23,27]. In a case of inverse hyperbolic sine (IHS) the transformation of the dependent variable is expected to produce consistent parameter estimates for the model in the presence of non-normality. The double-hurdle model with specification adjustments for heteroscedasticity and nonnormality is estimated for the smallholder farmer's household wealth on the adoption of GLOBAL GAP standards. Using maximum likelihood ratio test procedure in Gauss version 3.5 we reject the restricted model of homoscedasticity in favor of the alternative variance specification (see Table A1). Also, the likelihood ratio tests also unanimously reject the normality restriction in favor of inverse hyperbolic sine (IHS).

In the application of the model, farmers are assumed to make two critical decisions; (i) regarding willingness to adopt and (ii) the extent to which they are willing to implement GLOBAL GAP. Each of the two hurdles is conditioned by the household's wealth index, both farm, and farmers' characteristics. Different latent variables are used to model each decision process in the double-hurdle model, with the probit model determining the probability that a farmer is willing to participate in GLOBAL GAP certification while the Tobit model determines the extent which farmers are willing to comply with GAPs standards. The model can be specified as:

1. $y_{i1}^* = w_i' \propto + \mu_i$ Decision to obtain GLOBAL GAPs certification
2. $y_{i2}^* = x_i'\beta \propto + \mu_i$ The extent of adoption of GLOBAL GAP standards

$$y_{i1}^* = x_i'\beta \propto + \text{ if } y_{i1}^* > 0 \text{ and } y_{i2}^* > 0 \tag{5}$$

where $y_{i1}^*$ is denoted as latent variable describing farmers willingness to acquire GLOBAL GAP certification while $y_{i2}^*$ is a latent variable describing the extent of GLOBAL GAPs adoption (size of land farmers applies GLOBAL GAP standards) and $y_{i1}^*$ is the observe d area that GLOBAL GAP is applied or (dependent variable) $\mu_i$ represent the error terms distributed as $\mu_i \sim N(0,1)$ and $\mu_i \sim N(0,\delta^2)$. In such case Yen and Jones [29] recommend allowing for heteroscedasticity that can be estimated using maximum likelihood expressed as;

$$
\begin{aligned}
L(\propto, \beta, h, 0) = {}& \prod_0 \left[ 1 - \varnothing(w_i' \propto)\varnothing\left(\frac{x'}{\delta_i}\right) \right] \\
& \times \prod_1 \left[ (1 + \theta^2 y_i^2)^{-\frac{1}{2}} \varnothing(w_i' \propto) \propto_1^{-1} \varnothing\left(\frac{T(\theta y_1)x_i'\beta}{\sigma_i}\right) \right]
\end{aligned}
\tag{6}
$$

To facilitate the assessment of the impact of repressors directly, we calculate marginal effects to provide a better understanding of the magnitudes of the extent of adoption of GAPs as recommended by Jensen and Yen (1996). This is expressed as:

$$E(y_i|y_i > 0) = \varnothing\left(\frac{x_i\beta}{\sigma_i}\right)^{-1} \int_0^\infty \left( \frac{y_i}{\sigma_i\sqrt{1 + \theta^2 y_i^2}} \varnothing\left(\frac{T(\theta y_1)x_i'\beta}{\sigma_i}\right) \right) \tag{7}$$

## 3. Results and Discussion

### 3.1. Descriptive Statistics of Households

Computation Wealth Index by the PCA

As previously indicated Principle Component Analysis technique is used in the study to compute farmer's wealth indices. As suggested by Vyas and Kumaranayake [30] all the variables were dichotomized (1 = yes 0 = No) to show the ownership of each household asset. The weights (effectively defined by factor scores) for each asset are computed separately for the well-endowed and poor farmers. Also, the wealth index takes into account the distribution of assets between the well-endowed and poor farmers a reflection of snap bean farmer's economic conditions. Table 1 summarizes the results on Principle Component Analysis of 12 combined assets indicators considered to be important in defining the wealth status of snap bean farmers. Based on the Kaiser criterion of using a variable with an eigenvalue greater than one, only the first five assets under the physical capital category are significant. Also, the first five components explain 59% of the variation. The top components are, namely; agriculture assets, livestock assets, productive durables, dwelling assets, and GLOBAL GAP assets (Table 1). Agricultural asset describes 21% of the total variances and was used to construct indicators for all assets that gave a positive weight.

**Table 1.** Principle Component Analysis Extraction.

| Component/Variables | Initial Eigen Values | | Scoring Factor | Cumulative Pro |
|---|---|---|---|---|
| *Physical capital* | Total | Variance | | |
| Agricultural assets | 2.625 | 1.374 | 0.218 | 0.218 |
| Livestock assets | 1.25135 | 0.081 | 0.104 | 0.323 |
| Productive durables | 1.16979 | 0.074 | 0.097 | 0.420 |
| Dwelling assets | 1.09541 | 0.092 | 0.091 | 0.511 |
| GLOBAL GAP related assets | 1.0028 | 0.078 | 0.083 | 0.595 |
| Consumer durables | 0.4013 1 | 0.081 | 0.033 | 1.000 |
| *Natural Capital* | | | | |
| Total farm | 0.779 | 0.098 | 0.065 | 0.805 |
| *Financial capital* | | | | |
| Access to credit | 0.821 | 0.041 | 0.068 | 0.740 |
| *Human Capital* | | | | |
| Labor capacity | 0.673 | 0.100 | 0.056 | 0.918 |
| *Social capital* | | | | |
| group membership | 0.681 | 0.007 | 0.056 | 0.862 |
| GLOBAL GAP subsidy support | 0.572 | 0.171 | 0.047 | 0.966 |

Source: Authors' survey, 2017.

Figure 2 presents the computed wealth index (WI) results used to distinguish the characteristics of the well endowed and poorly endowed households. The method is recommended when categorizing households WI with 95% confidence level. The mean for poorly endowed households is −0.977, while the mean sample for well-endowed households is 1.029. Further, the findings presented in Table 2 indicate that 49% of the households are categorized as wealthy, and 51% as poorly endowed.

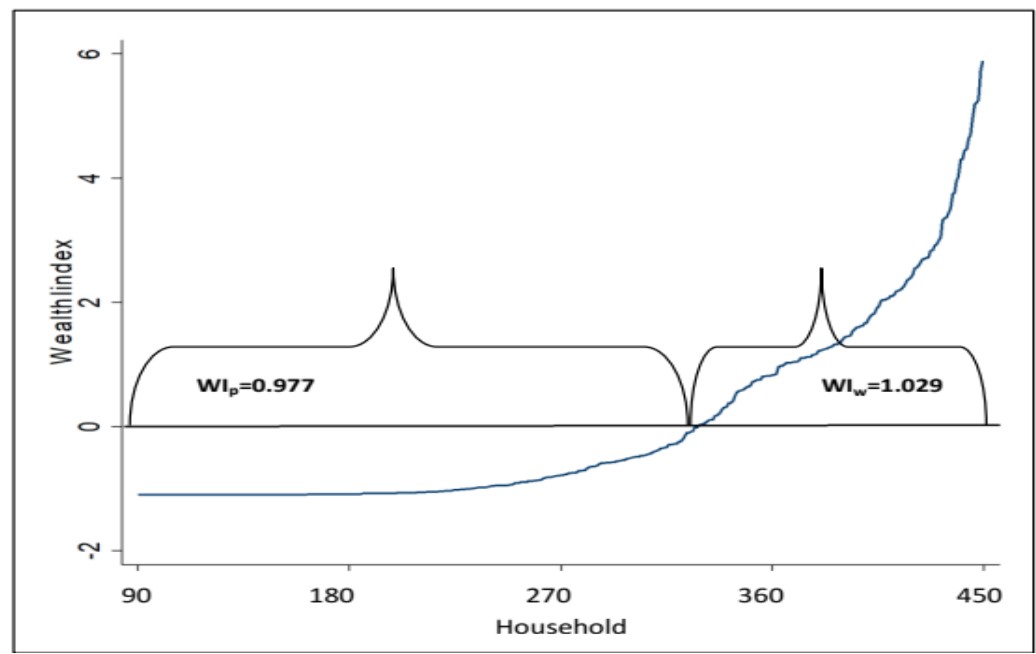

**Figure 2.** Sample Mean of the Standardized Wealth Index. Source: Authors' survey, 2017.

**Table 2.** Household Wealth indicators by Wealth Category.

| Combined Variables | Well Endowed (n = 211) | | Poor Endowed (n = 218) | |
|---|---|---|---|---|
| | Mean | Factor Score | Mean | Factor Score |
| Agricultural assets | 13.17% | 0.637 | 28.1% | 0.588 |
| Livestock assets | 26.3% | 0.554 | 13.7% | 0.521 |
| Productive durables | 35.6% | 0.611 | 14.5% | 0.531 |
| Dwelling assets | 17.4% | 0.571 | 10.2% | 0.091 |
| GLOBAL GAP related assets | 32.1% | 0.078 | 26.3% | 0.471 |
| Consumer durables | 65.2% | 0.595 | 23.5% | 0.594 |
| Total farm | 10.6% | 0.597 | 1.8% | 0.504 |
| Access to credit | 4.6% | 0.462 | 2% | 0.512 |
| Labor capacity | 11.1% | 0.509 | 9.1% | 0.501 |
| Membership to GLOBAL GAP farmers groups | 7% | 0.566 | 6.6% | 0.408 |
| GLOBAL GAP Subsidy support | 5% | 0.565 | 4% | 0.499 |

Source: Authors' survey, 2017.

The factor score in Table 2 shows the relationship between asset weights and classification status. Generally, the results reveal that the most valuable assets owned by all farmers include; livestock assets, consumer durables, agricultural assets, dwelling assets, and productive assets. We also observe that 32% of the well-endowed farmers invested in GLOBAL GAP assets while, in comparison, only 26% of the poorly endowed farmers invested in GLOBAL GAP assets. Further results reveal that membership to GLOBAL GAP affiliated groups, access to agricultural credit and GLOBAL GAP subsidy support benefits are the least likely resources that directly contribute to the wealth of households. Nevertheless, it is important to point out that the four factors can be considered to be exogenous; most often, they facilitate access to livelihood portfolio assets and resources at the household in the long run [12].

Figure 3 shows the average monthly proportion expenditure between the well-endowed and poorly endowed households. The well-endowed households slightly spend Ksh 1250 much of their income on None-staple fresh food than poorly endowed households. Also, we note that in comparison to the poor households, the well-endowed farmers invest more on-farm inputs and less on none staple foods. In discussing farm household economics Genius, Koundouri [15] observe that farmers are most likely to engage in capital-intensive ventures if it supposedly leads to improved yields and incomes.

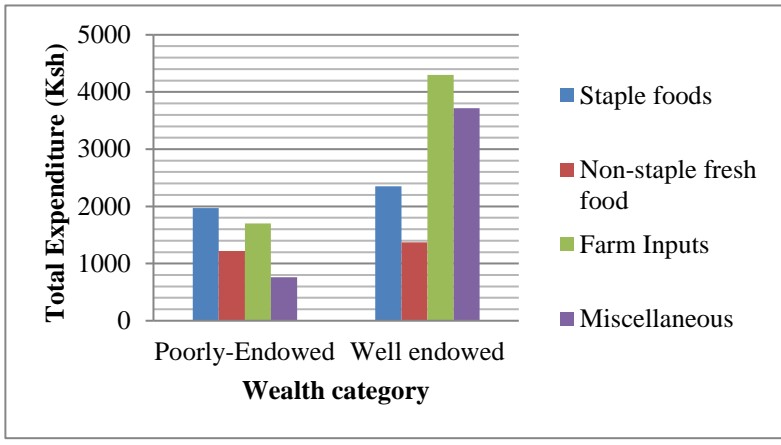

**Figure 3.** Household Expenditure Profile. Source: Authors' survey, 2017.

*3.2. Distribution of Wealth by Groups*

The results in Table 3 show the socio-demographic mean values between well endowed and poorly endowed farmers. The table also shows GLOBAL GAP certification adoption levels between the two categories of farmers. Generally, we observe that well-endowed farmers with GLOBAL GAP certification were older and had better returns from snap beans. Interestingly we observe that participation in GLOBAL GAP subsidy support programs is significantly correlated to GLOBAL GAPs certification for both poor and well-endowed GLOBAL GAPs certified farmers. Compared to poorly endowed farmers, well-endowed farmers accessed credit support and, in particular, farmers who are GLOBAL GAP certified. The results agree with Mohamed and Temu [13] study that assets accumulation guarantees financial credit access in cases where the collateral is required. Further findings reveal that wealthy households acquired invested more in GLOBAL GAP assets and gained higher returns than poorly endowed households. While farmers who acquire GLOBAL GAPs certification produced under marketing contracts and participate in GLOBAL GAP affiliated farmers groups more than None-GLOBAL GAP certified farmers. To supplement the farm income, more than 50 percent of the farmers earned income besides farming snap bean. However, we note that the well-endowed farmers engage in off-farm income-earning activities more than the poorly endowed farmers. The findings contrast Reardon, Berdegué [31] findings that poor farmers are likely to have off-farm income than wealthy farmers.

**Table 3.** Descriptive Statistics of Variables Included in the Estimations.

| | Poor Endowed (n = 229) | | | | | | Well Endowed (n = 221) | | | | | |
|---|---|---|---|---|---|---|---|---|---|---|---|---|
| | GLOBAL GAP Certification | | None GAP Certification | | t-Test | | GLOBAL GAP Certification | | None GAP Certification | | t-Test | |
| | Mean | Std. Dev. | Mean | Std. Dev. | t | p | Mean | Std. Dev. | Mean | Std. Dev. | t | p |
| Age of household head | 43.64 | 14.53 | 42.59 | 12.60 | −0.558 | 0.288 | 47.80 | 11.98 | 40.43 | 10.89 | −3.593 *** | 0.000 |
| Education years of household head | 9.810 | 2.567 | 9.410 | 2.893 | −1.021 | 0.154 | 9.723 | 2.320 | 10.04 | 2.489 | 0.794 | 0.786 |
| Cultivated land size (ha) | 1.042 | 1.026 | 1.450 | 1.191 | 2.553 | 0.994 | 1.107 | 0.993 | 1.375 | 1.121 | 1.510 | 0.933 |
| Off farm income,1 if yes 0 otherwise | 0.784 | 0.413 | 0.604 | 0.490 | −2.758 *** | 0.003 | 0.705 | 0.456 | 0.609 | 0.493 | −1.190 | 0.117 |
| Access to credit, 1 if yes 0 otherwise | 0.316 | 0.468 | 0.172 | 0.379 | −2.467 *** | 0.007 | 0.260 | 0.440 | 0.268 | 0.448 | 0.103 | 0.541 |
| GLOBAL GAP Subsidy support, 1 if yes 0 otherwise | 0.582 | 0.496 | 0.316 | 0.466 | −3.948 *** | 0.000 | 0.573 | 0.495 | 0.365 | 0.487 | −2.418 *** | 0.008 |
| GLOBAL GAP training, 1 if yes 0 otherwise | 0.506 | 0.503 | 0.345 | 0.477 | −2.333 ** | 0.010 | 0.505 | 0.501 | 0.341 | 0.480 | −1.899 ** | 0.029 |
| Contract farming, 1 if yes 0 otherwise | 0.734 | 0.444 | 0.302 | 0.460 | −7.413 *** | 0.000 | 0.705 | 0.460 | 0.292 | 0.460 | −5.466 *** | 0.000 |
| Membership to GLOBAL GAP groups, 1 if yes 0 otherwise | 0.848 | 0.361 | 0.637 | 0.482 | −3.372 *** | 0.000 | 0.840 | 0.367 | 0.725 | 0.452 | −1.702 ** | 0.045 |
| Distance to market (KM) | 4.537 | 3.615 | 4.561 | 3.345 | 0.047 | 0.519 | 4.147 | 3.318 | 4.425 | 2.949 | 0.486 | 0.686 |
| Snap bean output Kgs | 1498 | 3390 | 921.1 | 1878 | −1.614 ** | 0.054 | 1481 | 2708 | 602.41 | 630.7 | −2.061 ** | 0.020 |
| Value Snap bean Sold Ksh | 57887 | 8272 | 25697 | 2977 | −4.318 *** | 0.000 | 76578 | 1426 | 20302 | 1711 | −2.730 *** | 0.003 |
| Wealth Index | −0.893 | 0.536 | −1.016 | 0.314 | −2.136 ** | 0.016 | 2.4405 | 1.128 | 2.164 | 1.001 | −1.433 * | 0.076 |

Note: ***, **, * Significant at 1, 5, 10, percent levels respectively Source: Authors' survey, 2017.

### 3.2.1. Adoption of GLOBALGAP Certification

Table 4 shows the estimated coefficients and standard errors between the well and poor endowed households. The empirical findings on the willingness to adopt GLOBAL GAP certification reveal that membership to GLOBAL GAP affiliated farmers groups positively and significantly influence on wealthier farmers to attain GLOBAL GAP certification. Through collective action, farmers groups can facilitate joint investments hence reducing the cost of investing in GLOBAL GAP assets. The finding is consistent with Ndegwa, Muthoka [32] observations that smallholder farmers can take advantage of economies of scale and reduce per farmer costs when farmers groups invest in high-end mechanized technologies. In general, the availability of GLOBAL GAP information through training significantly influences GAP certification for all types of farming households. Humphrey [1] shows that inadequate training on food safety production makes it hard for smallholder farmers to acquire food certification and produce at the set standards. Land size would be a considerable factor that negatively influences wealthy farmers to invest in GLOBAL GAP certification. A similar observation also shared by Xiang, Huang [33]. The well-endowed farmers have a relatively positive and significantly better wealth index, an indication that they can financially invest in GAP certification. Interestingly [34] predicted that smallholder farmers are likely to abandon high-value export production due to low investments and high cost of compliance with food safety standards. Further results reveal that off farm income was a significant factor that would influence GAP certification for well and poor endowed farmers.

The total returns from snap beans significantly influence both categories of farmers to attain GLOBAL GAP certification. The results also reveal that irrespective of the wealth status, availability of marketing contracts significantly impact on the adoption of GLOBAL GAP certification. Further findings show that the number of years of GLOBAL GAP certification does play a significant role in influencing certification, particularly for the well-endowed and poor farmers. While selling snap beans to GAP certified buyer would significantly influence well-endowed to adopt GLOBAL GAP certification more than the poorly endowed.

**Table 4.** Hurdle I: Probability of Adopting *(dichotomous)*.

|  | Well Endowed (n = 221) | | Poorly-Endowed (n = 229) | |
| --- | --- | --- | --- | --- |
|  | Coeff | Std-Err | Coeff | Std-Err |
| Age of the household head | 0.015 | 0.005 | 0.015 ** | 0.004 |
| Education years of the household head | 0.042 * | 0.025 | 0.009 | 0.076 |
| Cultivated land size (Ha) | 0.415 | 0.102 | −0.236 * | 0.711 |
| Off farm income, 1 if yes 0 otherwise | 0.932 ** | 0.264 | 0.253 | 0.239 |
| Access to credit, 1 if yes 0 otherwise | 0.176 | 0.140 | −0.701 | 0.212 |
| GLOBAL GAP Subsidy support, 1 if yes 0 otherwise | 0.189 | 0.159 | −0.346 | 0.244 |
| GLOBAL GAP trainings, 1 if yes 0 otherwise | 0.170 *** | 0.152 | 3.327 *** | 0.700 |
| Contract farming, 1 if yes 0 otherwise | 0.106 *** | 0.140 | 1.175 *** | 0.331 |
| Membership to GLOBAL GAP farmers groups, 1 if yes 0 otherwise | 1.863 ** | 0.260 | −0.028 | 0.337 |
| Distance to market (KM) | −0.009 | 0.023 | −0.078 | 0.041 |
| Wealth Index | 0.1526 * | 0.509 | 0.116 | 0.119 |
| Snap bean returns | 1.210 *** | 0.0.2 | 6.250 * | 3.190 |
| Years of GAP certification | 0.072 *** | 0.018 | 0.107 *** | 0.024 |
| Snap bean output | 0.006 *** | 0.002 | −0.020 | 0.002 |
| GAP certified buyer, 1 if yes 0 otherwise | 1.739 *** | 0.283 | 0.332 *** | 0.427 |
| cons | −3.394 | 1.015 | −3.14 | 1.049 |

**Note:** \*\*\*, \*\*, \* Significant at 1, 5, 10, percent levels respectively. Source: Authors' survey, 2017.

### 3.2.2. The Extent of GLOBAL GAP Adoption

The results of the second hurdle estimates on the extent of GLOBAL GAP adoption are presented in Table 5. The dependent variable is the percentage area of snap bean produced in full compliance to GLOBAL GAP standards. Generally, the results indicate that not all determinant factors that influenced

adoption GLOBAL GAP certification would influence on the compliance to GAP standards during production. For instance, we observe that producing snap beans with marketing contracts significantly influence strict adherence to GLOBAL GAPs standards for all classes farmers. Likewise, selling beans to GAP certified buyers significantly enables farmers to comply to the expected standards. Otherwise, low compliance levels to GLOBAL GAP standards would increase the risk of snap bean rejection by buyers. The study findings further reveal that snap bean returns influence well-endowed farmers to meet the required standards ensures farmer's higher returns.

**Table 5.** Hurdle II Extent GAP of Adoption.

| | Well Endowed (n = 221) | | Poorly Endowed (n = 229) | |
|---|---|---|---|---|
| | **Coeff** | **Std-Err** | **Coeff** | **Std-Err** |
| Age of the household head | −0.012 | 0.012 | −0.011 | 0.028 |
| Education years of the household head | 0.098 | 0.051 | 0.036 | 0.042 |
| Cultivated land size (Ha) | 0.335 ** | 0130 | 0.025 | 0.11 |
| Off farm income, 1 if yes 0 otherwise | 0.670 *** | 0.249 | 0.211 * | 0.23 |
| Access to credit, 1 if yes 0 otherwise | −0.089 | 0.27 | 0.205 | 0.25 |
| GLOBAL GAP Subsidy support, 1 if yes 0 otherwise | 0.480 ** | 0.241 | 0.402 | 0.242 |
| GLOBAL GAP trainings, 1 if yes 0 otherwise | 0.238 | 0.242 | −0.286 | 0.206 |
| Contract farming, 1 if yes 0 otherwise | 0.666 ** | 0.244 | 0.384 ** | 0.234 |
| Membership to GLOBAL GAP farmers groups, 1 if yes 0 otherwise | 0.776 * | 0.381 | 1.440 | 1.359 |
| Distance to market (KM) | 0.061 | 0.411 | 0.028 | 0.080 |
| Wealth Index | 0.335 ** | 0.141 | 0.067 | 0.096 |
| Snap bean returns | 0.002 * | 0.123 | −0.002 | 0.003 |
| Years of GLOBAL GAP certification | 0.037 *** | 0.012 | 0.005 | 0.009 |
| Snap bean Output | 0.002 | 0.009 | -0.002 | 0.003 |
| GAP certified buyer, 1 if yes 0 otherwise | 0.776 *** | 0.426 | 0.686 ** | 0.223 |
| Cons | 1.985 | 0.603 | 3.06 | 0.648 |
| Log-likelihood | −403.2 | | −99.55 | |
| Pseudo $R^2$ | 0.055 | | 0.336 | |
| Prob > chi$^2$ | | | 0.000 | |
| lnsigma | −0.384 *** | 0.081 | −0.508 *** | 0.154 |
| /sigma | 0.680 | 0.055 | 0.601 | 0.601 |

**Note:** *** , ** , * Significant at 1, 5, 10, percent levels respectively. Source: Authors' survey, 2017.

The results are shown in Table 5 also demonstrate that participation in GLOBAL GAP farmers group positively effects on well-endowed farmers to keenly observe GLOBAL GAP standards more than the poorly endowed farmers. Interestingly we note that subsidy support only facilitates well-endowed farmers to comply with GAP standards. As pointed out by Luvai [35], even after subsidizing approximately US$ 6,500 to 30 farmers to attain GAP certification successfully, farmers still grappled with challenges of inadequate resources and capacity to bear the costs associated with compliance. The empirical estimate reveals that compliance to GLOBAL GAP not likely to be significantly influenced by household assets (wealth index) of farmers. The finding by Twine, Rao [36] found that liquidity-constrained farmers are less likely to acquire new crossbreeding technology in Tanzania

The parameter estimates of marginal effects derived from the hurdle model are presented in Table 6. Considering that most farmers produce snap bean under own irrigation blocks of plots ranging from 0.15 to 1.5 hectors, we note land size was a factor that would lower GLOBAL GAP compliance levels, especially for the well-endowed farmers. As Fintrac [37] observed, sometimes producing snap beans in smaller plots facilitates better management, regarding labor and irrigation systems. The snap bean output also reduced levels of GAP compliance for the poorly endowed more than well endowed. On average, the cost of compliance is considered to be at 9.51 US$/ton/year or 3.8% of the product price and ranging between 0.3% and 15.2% between the year [38]. The results on marginal effects presented in Table 6 show that the land size cultivated would reduce the compliance of GLOBAL GAP s by 10% for the well-endowed farming households. The GLOBAL GAP subsidy support was likely to

influence well-endowed farmers to comply to GLOBAL GAP standards by 48% and poorly-endowed by 40%. The results also show that selling snap beans GAP buyers increased well-endowed farmer's compliance level by 73% and 68% for the well-endowed farmers.

**Table 6.** Marginal Effects.

| Variables | Well Endowed (n = 221) | | Poorly Endowed (n = 229) | |
|---|---|---|---|---|
| | Marginal Effects dy/dx | | | |
| Age of the household head | 0.006 | 0.008 | −0.009 | 0.008 |
| Education of the household head | −0.031 | 0.041 | 0.068 | 0.042 |
| Cultivated land size (Ha) | −0.103 * | 0.107 | 0.025 | 0.109 |
| Off farm income, 1 if yes 0 otherwise | 0.669 *** | 0.248 | 0.211 | 0.229 |
| Access to credit, 1 if yes 0 otherwise | −0.088 | 0.269 | 0.205 | 0.249 |
| GLOBAL GAP subsidy support, 1 if yes 0 otherwise | 0.480 ** | 0.240 | 0.402 * | 0.241 |
| GLOBAL GAP trainings, 1 if yes 0 otherwise | 0.238 | 0.241 | −0.285 | 0.206 |
| Contract farming 1 if yes 0 otherwise | 0.665 ** | 0.244 | 0.383 | 0.233 |
| Membership to GLOBAL GAP farmers groups, 1 if yes 0 otherwise | 0.002 ** | 0.267 | −0.223 | 0.243 |
| Distance to market (KM) | −0.003 | 0.031 | −0.053 | 0.036 |
| Wealth Index | 0.334 ** | 0.141 | 0.066 | 0.095 |
| Snap bean returns | 3.270 | 2.600 | 2.680 * | 1.390 |
| Years of GAP certification | 0.037 *** | 0.011 | 0.005 | 0.008 |
| Snap bean output | 0.001 | 0.007 | −0.356 * | 7.205 |
| Gap Buyer, 1 if yes 0 otherwise | 0.738 ** | 0.258 | 0.685 ** | 0.223 |

**Note:** ***, **, * Significant at 1, 5, 10, percent levels respectively. Source: Authors' survey, 2017.

## 4. Conclusions

This study contributes to the literature on the impact of household wealth on the adoption of food certification standards as well as compliance with the standards during the production of snap beans. Specifically, we used the Principal Component Analysis technique to establish a farmer's essential assets and compute a farmer's wealth indices. We established that the most valuable assets that determine the wealth status of the study farming community include livestock assets, consumer durables, agricultural assets, dwelling assets, and productive assets. Also, the well-endowed farmers were more likely to invest in GLABAL GAP assets than the poor endowed farmers. Henson, Masakure [39] pointed out that assets such as generators, grading shades, sprayers, storage crates easily facilitated farmers to comply with GLOBAL GAP food safety standards. Similarly, when discussing livelihood strategies of farmers, Langyintuo and Mungoma [14] note that farmers invest in farm assets based on the need to intensify agricultural production.

In the study, we use the double hurdle model to establish factors that influence the adoption of GLOBAL GAP certification and the extent of the adoption between the well-endowed and poor endowed farming households. First, we establish that subsidy support was an important factor that influenced GLOBAL GAP certification. However, it is the wealthy farmers benefited more from the GLOBAL GAP certification. From a policy standpoint, it may be beneficial for subsidy support providers to consider the wealth status when providing farmers assistance. We also report that membership to GLOBAL GAP farmers groups was a significant factor that influenced GAP certification for the well-endowed farmers. As previous literature reports farmers, associations can easily establish a base of co-financing or register to become legal entities that enable members to purchase inputs, seek to produce marketing contracts, and borrow money [40].

The study shows that GLOBAL GAP training was a positive significant factor that influences farmers to certification for wealthy farmers. However, GLOBAL GAP training does not have a meaningful impact on extent standards implementation GAP standards irrespective of wealth status. Asfaw, Mithöfer [41] study report that delivery of information to a large number of smallholder farmers sometimes living in inaccessible areas is a big challenge. Alternatively, most farmers in the study area used television, radio, and mobile phones as a mode of receiving extension information; the trend is

becoming popular among households in emerging economies [42]. Irrespective of the wealth status, we observe that contract farming increased farmers' willingness to get GLOBAL GAP certification and also implement GAP production standards in the farms. This is an indication that with the readily available market for their produce, smallholder farmers with contracts are optimistic and positive to meet the set GLOBAL GAP standards. This is in agreement with Dedehouanou, Swinnen [43] that contract farming is considered to have prospects to change the livelihoods of the smallholder farmers despite the majority of them being suspicious that contracts favor agribusiness firms.

In conclusion, our findings have two policy implications. First, we observe that the wealth status of the household influence farmers to invest in GLOBAL GAP certification, and also enables adherence to production within GAP standards. Indeed, the cost of complying with GLOBAL GAP is too high and prohibitive to the poor farmers who majorly are not able to access financial credit. Hence to promote snap bean production, agencies should increase the support given to small-scale farmers, for instance, by facilitating access to credit, reduce the costs of compliance by providing subsidy inputs, especially to the poor households and capacity building on farmers to access crucial information, skills and training. While this study contributes to the discussion of food standard certification, much focus should consider establishing institutional and policy factors that affect food safety compliance in Sub-Saharan African countries.

**Author Contributions:** C.N.G.: Conception or design of the work, data collection, data analysis and interpretation results, dafting manuscript; J.H.: Critical revision of the article, funding acquisition, methodology; T.N.: Data curation Critical revision of the article. All authors have read and agreed to the published version of the manuscript.

**Funding:** This research was funded by the financial support of East Africa's Agricultural Policy and its impact on Sino-East Africa Agricultural Cooperation project and Priority Academic Program Development of Jiangsu Higher Education (PAPD). Grant no KYGB201802.

**Acknowledgments:** Sincerely gratitude to anonymous reviewers for the comments used to improve the manuscript and Egerton University students who were enumerators in collect data.

**Conflicts of Interest:** The authors declare no conflict of interest. The funders had no role in the design of the study; in the collection, analyses, or interpretation of data; in the writing of the manuscript, or in the decision to publish the results.

## Appendix A

**Table A1.** Maximum Likelihood Ratio Test.

|  | Likelihood Ration Test of Homoscedasticity Restriction | Likelihood Ratio Test of Normality Restriction |
|---|---|---|
|  | $H_0$ = homoscedasticity Error structure | $H_0$ = Untransformed dependent variable |
|  | $H_1$ = Heteroscedastic Error specification | $H_1$ = HIS Transformation to depend variable |
| No observations | 450 | 450 |
| Test statistic | 19.73 | 1.556 |
| Critical value | $X^2 = 0.1723$ Reject $H_0$ | $X^2 = 1.450$ Reject $H_0$ |

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
