# Peer review of "The Impact of Household Wealth on Adoption and Compliance to GLOBAL GAP Production Standards: Evidence from Smallholder farmers in Kenya"

_agriculture, doi:10.3390/agriculture10020050_

Round 1
Reviewer 1 Report
Comments
SUMMARY
The paper addresses the research area related to “food safety” of the MDPI Agriculture journal. I believe that the target journal is an appropriate forum for this article. The paper aims to empirically show the impact of smallholder farmer’s household wealth on adoption and compliance of food safety certification standards such as GLOBAL GAP.
BROAD COMMENT
The Introduction section is well done. The methodology is well written and detailed. However, when doing a survey, to ensure that the information gathered is representative and to provide ‘evidence’ of facts, certain rules in scope and methodology must be considered and observed. A key issue is to define the right sample size. In my opinion, the only shortage of this paper is related to the methodology section. The authors fail to conduct focus group discussions with the farmers in order to double-check the information gathered from the questionnaire survey (individually). Besides, another weakness of this study is that the authors failed to put the conclusion of the study in a big picture; it is too specific to the study area and the experiment. Please do include more implications of the results of the study in the conclusion section.
SPECIFIC COMMENTS
Line 75: How did you determine the sample size for the survey? Please, if you used a mathematical formula, include it in the manuscript.
Lines 124-145: Which econometric tests did you conduct to test the suitability of the data before applying the Probit and Tobit models? Do include the detail about those tests.
Lines 315-324: Include a separate section of conclusion in the manuscript. Please, do include more implications of the results of the study in the conclusion section.
Author Response
Response to Reviewer 1 Comments
Point 1: Line 75: How did you determine the sample size for the survey? Please, if you used a mathematical formula, include it in the manuscript.
Response 1:
- As advised, we have made the clearly given steps applied in sampling methodology of the study. Also, we have included the sampling formula used in the study. See section 2.1 sampling procedure.
Point 2: Lines 124-145: Which econometric tests did you conduct to test the suitability of the data before applying the Probit and Tobit models? Do include the detail about those tests.
Response 2:
To estimate the suitability of the model we have added more information on how we used maximum likelihood ratio test to for heteroscedasticity and inverse homoscedasticity hyperbolic sine (IHS) for dependent variables. We also present the results of the maximum likelihood ratio test in Appendix 1. See section 2.2.2 line 169 to 185 and Appendix 1
Point 3: Lines 315-324: Include a separate section of conclusion in the manuscript. Please, do include more implications of the results of the study in the conclusion section.
Response 3
- The conclusion section has been reorganized we have included discussions on results that are directly addressing the study objective in the section. See section 0

Reviewer 2 Report
Overall, I think it's a well-organized paper in setting topics, choosing research methods, and interpreting results.
I would like to suggest some minor comments.
First of all, a richer theoretical review of the relationship between farmer's wealth status and GLOBAL GAP acquisition will help us to understand why you have analyzed this subject.
Second, please check that equation (1) is correct.
Third, for Figure 3, I think the bar graph is more appropriate.
Author Response
Response to Reviewer 2 Comments
Point 1: First of all, a richer theoretical review of the relationship between farmer's wealth status and GLOBAL GAP acquisition will help us to understand why you have analysed this subject.
Response 1:
We have used relevant literature with theoretical backing to provide more insight on Global Gaps and wealth of farmers. See Section 1.0
We have slightly re-organized, the paper structure to facilitate easy understanding. See section 1.0
Point 2: Second, please check that equation (1) is correct
Response 2:
- The equation has been edited and arranged in systematic way to facilitate readers understand the information. See section 2.2.1 line 134 to 153
Point 3: Third, for Figure 3, I think the bar graph is more appropriate.
Response 3
As recommended, we have included bar graph in the manuscript. See section 3.11 figure 3

Round 2
Reviewer 1 Report
I have undertaken a review of the manuscript (revised) as well as the attached author responses to the initial review where I recommended major revisions. I am satisfied with the revisions made by the authors as they have addressed most, if not all, of my initial comments. Therefore, I do believe that the manuscript has been significantly improved and
